# Attention toward Social and Non-Social Stimuli in Preschool Children with Autism Spectrum Disorder: A Paired Preference Eye-Tracking Study

**DOI:** 10.3390/ijerph21040421

**Published:** 2024-03-30

**Authors:** Sara Congiu, Giuseppe Doneddu, Roberta Fadda

**Affiliations:** 1Department of Pedagogy, Psychology, Philosophy, University of Cagliari, 09126 Cagliari, Italy; robfadda@unica.it; 2Center for Autism and Related Disorder, Nuovo Centro Fisioterapico Sardo, 09131 Cagliari, Italy; io.donez@gmail.com

**Keywords:** autism spectrum disorder, preschool children, visual preference, eye-tracking, social stimuli, social attention, face processing

## Abstract

Different dimensions of visual attention to social (human faces) and non-social stimuli (objects) were assessed in 19 preschool children with Autism Spectrum Disorder (ASD) and 19 typically developing (TD) age, gender, and IQ-matched controls through an original paired preference eye-tracking paradigm. The present study found a significantly reduced attentional bias toward human faces in children with ASD compared to TD controls. The analysis of the total fixation time showed a significantly reduced preference for faces in children with ASD compared to TD children. Moreover, while TD children showed a significant preference for the face over the object, children in the ASD group observed the two paired pictures for a similar amount of time, thus showing no preference. Besides, children with ASD paid significantly more sustained attention to the objects than TD children. Children in the TD group paid greater sustained attention to the faces over the objects, while children in the ASD group did not differentiate between objects and faces. Finally, an age effect was found in ASD, as younger children in the group tended to prefer objects and to show more sustained attention towards them. Overall, these findings add to the literature on anomalies in attention toward social and non-social stimuli in young children with ASD compared to their TD counterparts. These results are discussed in the light of previous studies and suggest possible directions for future research.

## 1. Introduction

While typical development social stimuli, such as faces [1] and direct eye-gaze [2], will attract infants’ attentions from the first hours of life, a low interest in social aspects of the environment is considered to be an early sign of the anomalous development of social cognition in autism spectrum disorder (ASD). A meta-analysis reports that the corpus of studies in this field has found gaze differences in individuals with ASD during the observation of social and non-social stimuli. A discrimination problem emerges early in life, with less attention being paid to eye and whole face regions that are crucial for accurate social perception while greater attention is given to non-social elements [3]. In fact, children with in autism spectrum disorder (ASD) often show a reduced interest in social stimuli during the course of their first year of life ([4,5,6,7,8,9,10] for a review). Moreover, increased attention towards objects and peculiar exploratory behaviours of objects have been documented in this population [11,12,13], and both these characteristics are among the diagnostic criteria of ASD [14].

In their pioneering eye-tracking study, Klin and colleagues were the first to document that adult subjects with ASD were more likely to fixate on non-social rather than social elements present in short dynamic scenes [15]. This study opened different research questions which are still under investigation. First, researchers investigated whether low attention towards social stimuli was generally present in the ASD population at various ages (e.g., [8,16]) and also whether it was accompanied by a heightened preference for non-social stimuli (e.g., [17,18]). Finally, different studies tried to determine the presence of a causal link between these two behaviours. On one hand, the lack of interest in social aspects of the environment could be the result of a heightened interest in non-social elements, as hypothesized by Sasson and Touchstone [19]; on the other hand, the interest in non-social elements could be a by-product of a primary deficit in social attention, like predicted, for instance, by the social motivation theory [20]. However, great methodological heterogeneity among the studies does not allow a definitive conclusion about the nature of social orienting difficulties in this population [3].

The hypothesis that children and adults with ASD might show a tendency to prefer non-social rather than social stimuli, appearing in a static or dynamic complex scene, has been supported by many studies involving participants of various ages [10,21]. For instance, Riby & Hancock [16] showed that adolescents with ASD fixated on the non-social elements of the background more often than the faces depicted in photographic scenes. Nakano and colleagues [17] found that, during the observation of video scenes, children with ASD compared to typical controls were significantly more attracted by strings of letters appearing on the bottom of the screen than by a little girl appearing on the above part of the screen. Preferential looking paradigms, in which paired social and non-social stimuli are displayed side by side, were used with toddlers and infants in several studies [18,22,23,24] and provided results supporting the preference for non-social stimuli over a variety of social stimuli in very young children with ASD.

In general, eye tracking studies operationalize different dimensions of visual attention or indexes. The first measure is the prioritization, which is whether the observer orientates first towards the social or the non-social stimulus. The second measure is preference, which is usually a measure of the total looking time toward the social or the non-social stimulus. A third measure is the mean duration of visits, taken as a measure of sustained attention towards the social or the non-social stimulus. In preferential looking paradigms, it is considered a typical behaviour to orient first toward social stimuli and to show a longer looking time and longer visits towards them compared to non-social ones.

Klin and colleagues [18] used a preferential looking paradigm with point light displays. They found that two-years-old toddlers with ASD tended to look more at parts of the scene characterized by high levels of non-social audiovisual synchrony rather than showing a preference for the display of human biological motion, like matched TD controls did. Pierce et al. [22], using a preferential looking paradigm that compared social (kids doing yoga movements) and non-social (geometric patterns) dynamic scenes, found that preference for the non-social stimuli was a predictor of ASD in toddlers at risk aged 14 to 42 months. This group looked significantly more at the non-social pattern than the other two control groups composed, respectively, of typically developing controls and developmentally delayed controls. The same result was confirmed in a successive study [23] in which toddlers with ASD aged 12 to 49 months showed a strong preference for moving geometric images over social images. Rutherford [24] used a preference task with static images to measure spontaneous social orienting in young infants at high likelihood to develop ASD (infants with an older sibling already diagnosed with ASD) at the ages of 3 and 6 months. The findings showed that infants at risk had a significantly reduced preference (looking time) for social stimuli (faces rather than non-face stimuli obtained by scrambling the same face image in order to preserve spatial frequency) compared to controls. Moreover, the group difference increased from 3 to 6 months, suggesting the presence of a worsening process over time.

However, other results do not support the hypothesis of a generalized preference for non-social over social stimuli in ASD, and some studies failed to find reduced interest in social stimuli.

Elsabbagh and colleagues [25] tested a group of infants at high likelihood for autism and a low likelihood control group of infants on a “face pop-out” task at the ages of 7 and 14 months. The stimuli were images in which a face was included among an array of other objects. The results showed no significant difference between the groups, suggesting that infants who were later diagnosed with ASD were significantly more attracted by the face stimulus at the age of seven months, like matched typically developing infants. Another prospective study involving infants at high likelihood for ASD [8] demonstrated that a decreased looking time at social elements of a complex dynamic scene, depicting a person and different objects, was not reflected in an increased looking time at objects. Compared to the control groups, 6-month-old infants later diagnosed with ASD attended less to the social scene, and when they did look at the scene, they spent less time monitoring the actress in general and her face in particular. However, limited attention to the actress and her activities was not accompanied by enhanced attention to objects compared to controls.

Pictures of objects related to circumscribed interests were found to attract more the attention of children with ASD aged between 2 and 5 years compared to age matched typical controls [12]. A further eye-tracking study investigated the preference for social versus non social stimuli in preschoolers with ASD aged between 24 and 62 months in regard to static images [19]. The study found that the preference for faces presented together with objects was similar in children with ASD and TD except when the paired object belonged to categories such as trains, airplanes and other vehicles, which are often among children with ASD’s restricted interests. This finding suggests that social attention in preschool-aged children with ASD can vary also as a function of the salience of the competing non social object, and this is probably related to the pattern of repetitive behaviours characterizing children with ASD, particularly in the area of circumscribed interests [19].

A similar paradigm was used in another recent study by Vacas and colleagues [26] in a paired preference task that compared happy, angry, and neutral faces with two types of objects (related or not related to autism circumscribed interests). The results showed that, relative to TD children, children with ASD aged between 44 and 72 months showed reduced attention to faces and a visual preference for objects, regardless of the type of object.

However, other studies showed different results about the influence of circumscribed interests on the gaze behaviour of children with ASD. Ambarchi and colleagues [27] used a similar experimental paradigm in older children with ASD (3 to 12 years) and age-matched typical controls, with their results showing a reduction in either social and object attention in ASD independent of the presence of circumscribed interests. Another eye-tracking study involving school-age children and adolescents with ASD and TD (aged between 6 and 17 years) found that competing objects related to circumscribed interests influenced attention to faces in the TD group but not in the ASD group. The duration of gazes to faces was low in children with ASD independently from the nature of the competing object [28].

In summary, eye-tracking studies regarding visual preference to social vs. non-social stimuli in children with ASD showed mixed results. The most robust finding seems to be a reduced attention for social stimuli compared to controls, while a heightened preference for non social stimuli is not always present. The present study aimed to contribute to the literature by investigating possible group differences in the attention toward social and non-social stimuli in preschool children with ASD. The ASD group was compared to age and cognitive functioning matched controls. Differences were evaluated in terms of prioritization, preference, and sustained attention. The current study wanted to investigate whether those differences would emerge even in the absence of circumscribed interest objects, which might elicit a heightened attention in children with ASD. To achieve this aim, an original eye-tracking visual preference task, using static images of faces and common objects not belonging to circumscribed interests, was constructed.

Children with ASD, compared to controls, were expected to show longer latencies before fixating on face stimuli, thus indicating a lack of prioritization or a reduced attentional bias, a lack of preference (i.e., a shorter total fixation time), and reduced sustained attention (measured in terms of the mean duration of visits) to face stimuli. Moreover, within-group comparisons were expected to provide further information about a lack of preference and sustained attention for social versus non social stimuli in ASD, while the opposite was expected for control children. Attention towards non social stimuli was analyzed, but no explicit hypothesis about a heightened preference or attention toward this kind of stimuli was formulated given that, in previous literature, this finding is less robust. However, a heightened attention towards objects in our paradigm would show that this core characteristic emerges even in the presence of common everyday objects, going beyond the presence of circumscribed interests.

## 2. Materials and Methods

### 2.1. Participants

A total of 38 preschool children took part in the study: 19 children with ASD (15 males, mean age = 56.63 months, SD = 8.42 months, age range = 45–73 months), and 19 matched typically developing controls (14 males, mean age = 60.42 months, SD = 2.26 months, age range = 57–64 months). The two groups were matched for gender, age, and non-verbal IQ.

Gender matching was chosen since child males and females may show differences in circumscribed interest-related objects [12,19,29]. The children had also similar chronological ages in order to reduce the influence of age-related differences in object preferences and maturational changes in the visual system. Moreover, the children in both groups were assessed with the Leiter-R Vr battery [30] in order to ensure a similar level of non verbal cognitive functioning (TD mean IQ = 109, SD = 10.69, range = 90–126; ASD mean IQ = 110, SD = 13.38, range = 90–133). The IQ value was missing for 2 children in the control group who were not at school when the Leiter-R session took place. Typically developing children were recruited at a local preschool in Cagliari, while children with ASD were recruited at the Center for Pervasive Developmental Disorders, Azienda Ospedaliera Brotzu, Cagliari. All the children in the ASD group were diagnosed by expert clinicians according to DSM-5 criteria and ADOS observation [31]. Children were included in the ASD group according to ASD diagnosis, their NVIQ was in the normal range, which is equal to or above 85 points (1 SD below the mean), and they were preschoolers, aged between 42 and 78 months, with an absence of comorbid symptoms of other developmental conditions (e.g., ADHD). Children included in the control group had a NVIQ equal to or above 85 (1 SD below the mean), were preschoolers, aged between 42 and 78 months, and any preoccupation for signs of any developmental disorder was absent, as reported by parents and teachers.

### 2.2. Stimuli

The stimuli were twenty images, each 1024 × 381 pixels in size (23.9 × 9 degree of visual angle), as seen in Figure 1. Each stimulus image was composed of the following: a picture of a human face (five males and five females), with a neutral expression, paired with a picture of an object on a white background. Ten pictures of faces were selected (5 male and 5 females) as follows: 4 male pictures were taken from a database in which they were categorized as neutral (showing no emotional expression) [32]; the other 6 stimuli were original face pictures of young adults (5 females and 1 male) showing a neutral expression similar to that of the pictures taken from the database. The pictures represented neutral expressions, according to the opinion of various adult judges, who were consulted informally before the study. We also referred to the previous literature in order to exclude circumscribed interest objects (e.g., transportation vehicles, or mechanical instruments) and include non circumscribed interest objects (like clothing or household items) [12,19,27]. The non-social images represented common everyday objects, toys, and clothing items, (a dish, a cup, a football, a table tennis racket, a wooden cube, a skateboard, a wooden toy car, a toy drum, a shoe and a hut). Neutral human faces were chosen as prototypical social stimuli, as attention to the face is supposed to be driven by fundamental and rapid orienting mechanisms, which produce an attentional bias even toward static pictures [33].

The pictures of faces and objects were similar in size and counterbalanced for position (left vs. right), and thus each face and each object was displayed two times. All the stimuli were in grayscale to exclude the influence of color saliency on visual preference. None of the depicted objects were related to a particular restricted interest of any of the children, as reported by parents.

Paired images were separated by an empty area of 520 × 381 pixels (12.32 × 9 degrees of visual angle). Areas of interest were traced manually following the profile of the displayed face and object pictures. Two areas of interest for each image were considered: the area of the face and the area of the object. The areas of interest of objects measured 253 × 213 pixels on average (6 × 5 degrees of visual angle), and the areas of interest of faces measured 199 × 256 pixels on average (4.73 × 6 degrees of visual angle).

### 2.3. Apparatus and Procedure

Eye movements were recorded by means of a Tobii-T60 eye-tracker (Tobii Technology) at a rate of 60 Hz (one gaze point every 17 ms) with an accuracy of 0.5°. Each participant was positioned on a chair at a distance of about 65 cm from the 17” TFT monitor with a resolution of 1280 × 960 pixels.

The eye-tracker was operated by an experimenter using a laptop computer not visible to the child. Prior to the testing session, each young participant successfully completed a five-point calibration procedure. After the calibration, children were simply invited to sit still and watch the images appearing on the screen. Each image was displayed on the screen for 5 s and was preceded by a central attractor composed of a sound and a moving image lasting about 2 s and displayed on a black background. The whole testing procedure lasted about 4 min.

### 2.4. Data Coding and Analysis

Fixation data were calculated through the Tobii Studio Software 2.3 version, applying the Tobii fixation filter (distance 35 pixels, and velocity 35 pixels).

The following three different measures of visual attention toward social and non-social stimuli were analysed: time to first fixation, which is the latency of first fixation towards the social or non-social stimulus; total visit duration, which is the mean overall looking time to the social and non-social stimulus; and mean duration of visits towards the social and the non-social stimulus. A visit is the sum of all consecutive fixations within an Area of Interest in seconds.

#### 2.4.1. First Fixation

First, we coded whether the first fixation was directed toward the social or the non-social stimulus. A shorter time to first fixation towards an area of interest indicates that the first fixation was directed towards that area. For the statistical analysis, a first fixation toward the object (non-social) was coded as 1, while a fixation on the area of the face (social) was coded as −1 (dichotomous variable). Therefore, a positive number indicates a majority of first fixations toward the object (non-social stimulus), while a negative number indicates a prevalence of first fixations toward the social stimulus (face), and a 0 indicates a chance level of first fixations. When a child fixated on the face but never fixated on the object during the five seconds of exposition to the stimulus, or vice versa, prioritization of the fixated area of interest was coded. When a child failed to fixate on either the face or the object, the data were considered as missing and excluded from the analysis. This coding was aimed at understanding whether the children prioritized attention towards the social stimulus.

#### 2.4.2. Total Visit Duration

In order to evaluate visual preference, the analysis considered the proportion of the total amount of time spent on each area of interest. Preference was operationalized as a greater amount of time spent fixating a specific area of interest. The eye-tracking metric “total visit duration” which measures the total fixation time within an Area of Interest (in seconds) was used in order to measure overall preference.

#### 2.4.3. Mean Time per Visit

In order to evaluate sustained attention towards social and non-social stimuli, the mean duration of the visits on each area of interest was calculated.

## 3. Results

### 3.1. Statistical Analysis

Preliminary analysis showed that age and non verbal IQ were normally distributed, while gender was not. Between group analysis excluded significant differences in terms of gender (*U* = 171, *p* = 0.707), age (*t* (df) = 1.893 (20.59) *p* = 0.073) or non-verbal IQ level (*t* (df) = −0.244 (34), *p* = 0.808), Table 1.

#### 3.1.1. Analysis of First Fixation Data: Prioritization

In order to evaluate the prioritization of attention towards the face, a prioritization index was calculated, which is the frequency of first fixations towards the face divided by the number of valid trials. This coding was aimed at investigating the prioritization of the social stimulus. Preliminary analysis of the data showed a violation of the assumption of normal distribution. Thus, a non-parametric analysis for the between-group comparison was applied.

#### 3.1.2. Analysis of Total Looking Time: Preference

Preference was analyzed as the total looking time spent on the stimulus. A preliminary analysis showed that the data of the total visit duration on the whole stimulus image violated the assumption of the normal distribution; therefore, a Mann–Whitney test was applied in order to compare the two groups. The results showed a significant between-group difference *U* = 76, *p* = 0.002, *d* = 0.978, as children with ASD (M = 4.18, SD = 0.633) looked less at the whole stimulus than TD children (M = 4.66, SD = 0.279). In order to control for the difference in attention towards the whole stimulus, the proportion of looking time towards the area of interest of the face was calculated by dividing the looking time toward the specific area of interest by the total looking time toward the whole image displayed on the screen. Total looking time (total visit duration) in regard to the face and towards the object: preliminary analysis on the proportion of looking time towards the face and the object showed that the data were normally distributed.

#### 3.1.3. Analysis of Mean Visit Duration: Sustained Attention

Sustained attention was investigated through the analysis of the mean visit duration. A preliminary analysis showed that the data of the mean visit duration on the whole stimulus image violated the assumption of the normal distribution; therefore, a Mann–Whitney test was applied in order to compare the two groups. The results showed a significant between-group difference *U* = 81, *p* = 0.004, *d* = 0.969, as the mean duration of visits to the whole stimulus in children with ASD (M = 3.796, SD = 0.784) was inferior to that of TD children (M = 4.407, SD = 0.425). In order to control for this difference, the proportion of visit duration towards the area of interest of the face and of the object were calculated by dividing the mean fixation time toward the specific area of interest by the total looking time toward the whole image displayed on the screen. Analysis of the Mean duration of visits towards the face and towards the object: preliminary analysis on the proportion of duration of the visits toward the face and the object showed that the data were normally distributed.

#### 3.1.4. Correlation Analysis

Finally, the correlations between independent (age, gender, IQ) and dependent (proportion of total looking time on face/object and proportion of duration of fixations) variables were analyzed. Separate correlation matrices were calculated for the two groups (ASD and TD).

### 3.2. Results of First Fixation Data: Prioritization

The analysis of first fixations revealed that 95% of TD and 73% of ASD children directed their first fixation towards the area of the face (TD M = −0.553, SD = 0.302; ASD M = −0.216, SD = 0.307), as seen in Figure 2. Individual response patterns show that only children in the TD group significantly prioritized the face, binomial test, *p* < 0.001, while children in the autism group did not significantly prioritize the face above the object, binomial test, *p* = 0.064.

A Mann–Whitney U test revealed a significant between-group difference *U* = 66, *p* < 0.001, *r_rb_* = 0.634, as the TD group (M = 0.777, SD = 0.151) showed a significantly higher prioritization of the face stimuli over the objects than the ASD group (M = 0.61, SD = 0.155), Figure 2.

### 3.3. Results of Total Looking Time: Preference

An independent sample *t*-test showed a significant between-group difference in the proportion of total looking time towards the face *t* (36) = 2.384, *p* = 0.022, *d* = 0.774, as children with typical development as a group spent more time fixating on the face (M = 0.51, SD = 0.067) than children with Autism Spectrum Disorder (M = 0.44, SD = 0.107). Thus, children in the ASD group showed a statistically significant reduced attention towards the social stimulus compared to TD children. An independent sample *t*-test showed no significant between-group difference in the proportion of looking time towards the object *t* (36) = −0.635, *p* = 0.53, showing that the two groups of children observed the object for a similar amount of time (TD M = 0.362, SD = 0.055, ASD M = 0.379, SD = 0.102). Thus, children in the ASD group did not show a heightened attention towards the non-social stimulus compared to TD children.

Moreover, a paired sample *t*-test on the proportion of looking time analyzed whether each group showed a visual preference for one of the two paired pictures presented. Children in the TD group *t* (18) = 5.721, *p* < 0.001, *d* = 1.312 showed a significant preference for the face over the object, while children in the ASD group observed the two paired pictures for a similar amount of time *t* (18) = 1.56, *p* = 0.136, showing no preference for one or the other, as seen in Figure 3.

### 3.4. Results of Mean Visit Duration: Sustained Attention

An independent sample *t*-test showed no significant between-group difference in the proportion of the duration of visits towards the face *t* (36) = −0.901, *p* = 0.373, showing that the two groups of children observed the faces with a similar amount of sustained attention (TD M = 0.224, SD = 0.050; ASD M = 0.242, SD = 0.069).

An independent sample *t*-test showed a significant between-group difference in the proportion of the duration of visits towards the object *t* (36) = −2104, *p* = 0.042, showing that children with ASD paid more sustained attention to the objects than TD children (TD M = 0.183, SD = 0.037, ASD M = 0.223, SD = 0.073).

Finally, a paired sample *t*-test on the proportion of visit duration analyzed whether each group paid a greater sustained attention to one of the two paired pictures presented. Children in the TD group *t* (18) = 3.050, *p* = 0.007, *d* = 0.175 paid significantly more sustained attention to the face over the object, while children in the ASD group visited the two paired pictures for a similar amount of time *t* (18) = 0.764, *p* = 0.455, showing no difference between the two in terms of sustained attention, as seen in Figure 4.

### 3.5. Results of Correlation Analysis

The results indicate an age effect in the ASD group. Age was positively correlated with the proportion of total looking time on the face (*r* (17) = 0.512, *p* = 0.025) and negatively correlated with the proportion of total looking time on the object (*r* (17) = −0.468, *p* = 0.043), meaning that, in the ASD group, older children tended to look more at the face while younger children tended to look more at the object. Moreover, in the group of children with ASD, age was negatively correlated also with the proportion of mean duration of fixations to object stimuli *r* (17) = −0.681, *p* = 0.001, as younger children paid longer sustained attention towards them. No significant correlation emerged for the group of TD controls.

## 4. Discussion

The present preferential viewing paradigm investigated between-group differences in terms of prioritization, preference, and sustained attention towards social and non-social stimuli.

Visual patterns of children revealed that children in the ASD group were significantly less likely to orient first towards the social stimulus than TD children; therefore, children with ASD did not prioritize the face above the object, while children in the control group did. Children with ASD showed a reduced preference for faces compared to controls, paying significantly higher sustained attention towards object stimuli than controls. Within group differences, in terms of preference and sustained attention towards social versus non-social stimuli, showed that typically developing controls preferred faces above objects, paying more sustained attention to them, while Children in the ASD group did not differentiate between social and non social stimuli in any of the two measures. Moreover, an age effect emerged for the ASD group exclusively, in that younger children in the group tended to prefer and to display more sustained attention to object stimuli.

The first fixation is likely to reflect attention capture [19] and thus an automatic prioritization of the stimulus. In typical development, human faces and face-like stimuli are generally prioritized over non-face stimuli from infancy (see [34] for a review), while a reduction in such an attentional bias was found in toddlers [35] and preschoolers [19,26] with ASD. The attentional bias toward faces is crucial for social and communicative development, as well as for language acquisition [36]. Children with ASD show a reduced social attention from early development [5,16], and this is supposed to hinder their successive social and communicative development. These results confirm previous findings of a significantly reduced attentional bias towards faces in young children with ASD compared to matched typical controls [19,26,35].

The present study also investigated whether children with ASD show a preference towards a face when it is paired with a common object, similarly to what is expected in typically developing children. The results showed the presence of a difference between groups in the overall amount of time spent fixating on the face stimuli. Children with ASD showed a significantly reduced overall preference for the social stimulus compared to controls, which is compatible with previous findings [10]. When the overall amount of time dedicated to the object was analyzed, in order to investigate the presence of group differences in the preference for the non-social stimulus, no between-group difference emerged, as the two groups observed the object for a similar amount of time.

Moreover, while TD children showed a significant preference for the face over the object, children in the ASD group observed the two paired pictures for a similar amount of time, thus demonstrating no preference for one or the other. These data show that children with ASD did not show a heightened preference for objects compared to controls, nor a differentiation between face and object stimuli in terms of preference.

However, a significant group difference emerged in terms of sustained attention. Children with ASD provided significantly more sustained attention to the objects than TD children. Moreover, while children in the TD group, paid more sustained attention to the face over the object. Children in the ASD group did not differentiate between the two paired pictures in terms of sustained attention.

In summary, these results suggest that young children with ASD can show a reduced preference for social stimuli compared to TD controls in terms of prioritization and overall attention, even in the absence of a preference for paired non-social stimuli represented by common objects. However, the results showed that children with ASD did not differ from controls in terms of sustained attention to faces, although they devoted significantly more sustained attention to objects than their typical peers.

Our results have two main implications: First of all, they mean that, if the objects are common, not preferred, and are not related to circumscribed interests, they might not be as appealing for children with ASD in comparison to faces. Secondly, as we know that the experience with the objects might play a key role in orienting visual attention, a longer sustained attention might foster a preference for objects rather than to faces during development. If we apply this information to the home environment and/or to the school setting, parents and teachers should invest valuable time in exposing children with ASD to faces so that the human faces might become more relevant compared to everyday objects in those environments.

Some previous studies have suggested that children with ASD can pay more attention towards a variety of non-social stimuli as well as reduced attention to social stimuli, such as human figures, faces, and eyes [16,17,18,22]. However, this result was not confirmed by other studies, which found a reduced interest for social stimuli but not a higher interest for non-social ones [25] unless in particular cases [19].

One possible explanation of such dissociation can be that only particular non-social stimuli, i.e., stimuli with specific characteristics, interfere with social attention in children with autism and not in all of them.

Sasson and colleagues suggested that social attention can be modulated by the salience of the competing non-social stimulus, proposing that objects related to restricted interest can be more attractive than human faces for some young kids with ASD [19].

Furthermore, the preference for non-social over social stimuli might also be related to unusual interests in sensory aspects of the environment often characterizing individuals with ASD. Specific categories of non-social stimuli may be particularly salient and interfere with attention for social stimuli. Among these, it can be suggested that dynamic geometric patterns [22,23], or coincident light and sound [18], might capture attention because they provide an attractive sensory stimulation.

The current paradigm displayed pictures of common objects, and thus not objects related to restricted interest, nor dynamic stimuli, which produce a particular sensory stimulation; this would explain why it did not elicit a heightened interest in children with ASD. This result is also compatible with previous studies in which static pictures of common objects were shown to children with ASD [19,25,28].

Moreover, the study found a reduced attention to social stimuli in young children with ASD in the absence of a heightened interest in non-social stimuli (see also [8]). This strongly suggests that these two aspects might depend on different underlying mechanisms driving attention.

On one hand, the social motivation theory of ASD [20] predicts that a reduced interest in social stimuli, occurring early on in development, corresponds to a loss of social learning opportunities, which exerts a detrimental cascading effect on the functional specialization of specific brain networks involved in social cognition, leading to the social and communicative impairments typical of ASD [18,36,37,38,39]. This would determine the discrimination problems [3]. Also, the reduced interest in social stimuli can worsen over time due to a lack of learning opportunities.

According to the DSM-5 criteria for ASD, reduced attention for social stimuli belongs to the area of persistent deficits in social communication and social interaction. On the other hand, a heightened attention towards social stimuli might be related to the area of restricted, repetitive patterns of behavior, interests, or activities. This suggests a possible dissociation between attention towards social and non-social stimuli in ASD.

Finally, an association between age and total fixation time in the ASD group emerged, in that younger children tended to prefer, and paid higher sustained attention to, objects, while older children tended to prefer faces. This is interesting given the small sample size and the narrow age range of the subjects. On one hand such a result might be considered surprising since, for instance, a decline in attention to the eyes was found to emerge very early on in development, particularly during the first six months of life, in ASD [9]. On the other hand, such an age effect might be due to early intervention. However, since we did not collect data about treatment in our sample, this can only indicate that, in future studies, it would be interesting to consider a possible effect of intervention on social attention.

Some limitations of this study have to be considered. The size of the sample is quite small, although it is similar to that of previous studies in the literature [19,26]. The subjects are preschoolers and the chosen age range is quite narrow; therefore, the results of this study are comparable to those of previous studies investigating attention to social stimuli in ASD and in typical controls at this particular age. Another limitation is that, in the presented study, we were not able to investigate gender differences since the number of girls (four in the ASD groups and five in the TD groups) is not sufficient to calculate a possible gender effect. Unfortunately, we did not collect information about possible eating behavior in the ASD children of our sample. It is well-established in the literature that children with ASD could show some issues in regard to eating behavior, specifically in food selectivity [40]. Usually food refusal is based on texture, color, and shape qualities of the food. Possible aversive reaction to empty cups or dishes per se have not been investigated in previous studies, to the best of our knowledge. However, this is an interesting new direction of research that it would be worth investigating. Finally, we did not consider the effect of early intervention as an intervening variable in our study. In the future, it may be important to take into account the possible mediating effect of early intervention in modulating social attention in children with ASD.

## 5. Conclusions

The results of the present study add to the literature regarding reduced attention towards social stimuli in children with ASD. Young children with ASD can show a reduced preference for social stimuli compared to TD controls; however, this is not automatically reflected in a heightened interest in all categories of non-social ones. Our result is compatible with a recent study supporting the social motivation theory, as school-age children with ASD showed a reduced prioritization and preference for face stimuli independent of the fact that the paired object was one of circumscribed interest [28]. Nevertheless, children with ASD paid more sustained attention to the objects than TD controls. This subtle difference is likely to reflect a characteristic of the disorder emerging beyond the presence of circumscribed interests. Still, the specific characteristics of non-social stimuli related to the area of restricted, repetitive patterns of behavior, interests, or activities are likely to modulate social attention in ASD.

As early attention to environmental stimuli can determine different trajectories of neural and behavioral specialization, future research should investigate how attention is attracted by different kinds of non-social stimuli in young children with ASD in order to understand how they can interfere with attention towards social stimuli in this population.

## Figures and Tables

**Figure 1 ijerph-21-00421-f001:**
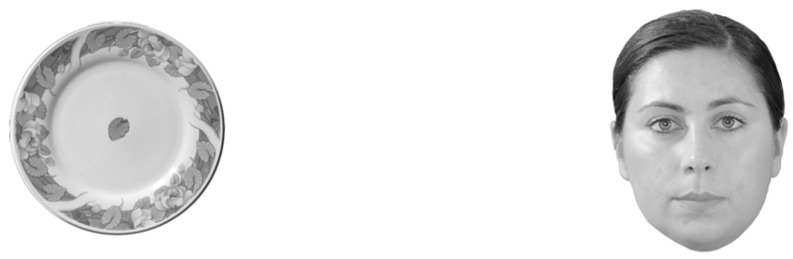
Example of stimulus image.

**Figure 2 ijerph-21-00421-f002:**
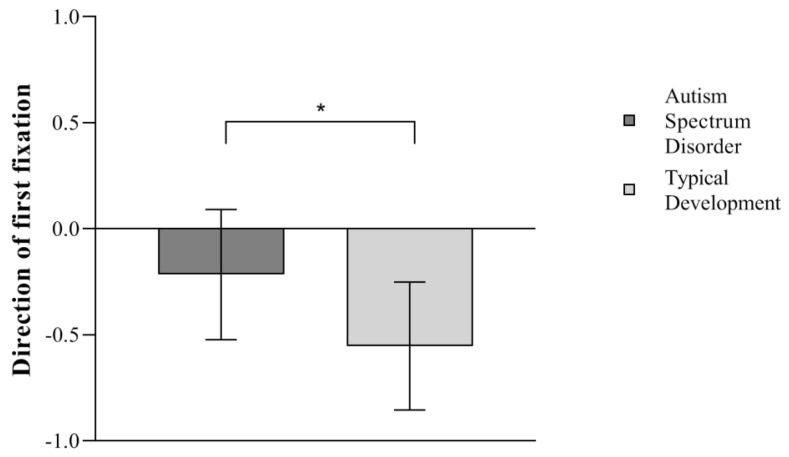
Mean First Fixation, * *p* < 0.05. Error bars depict a 95% confidence interval.

**Figure 3 ijerph-21-00421-f003:**
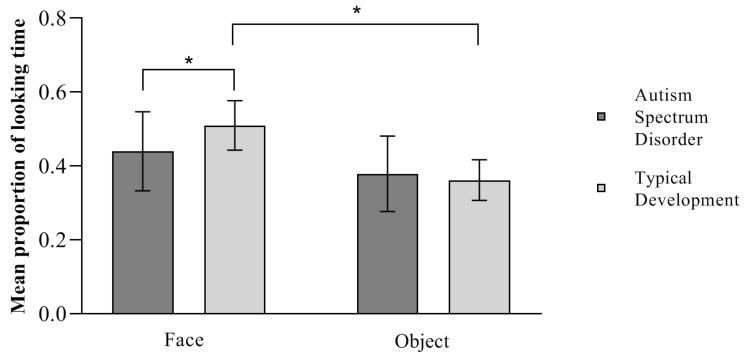
Mean proportion of looking time towards the stimuli. * *p* < 0.05. Error bars depict a 95% confidence interval.

**Figure 4 ijerph-21-00421-f004:**
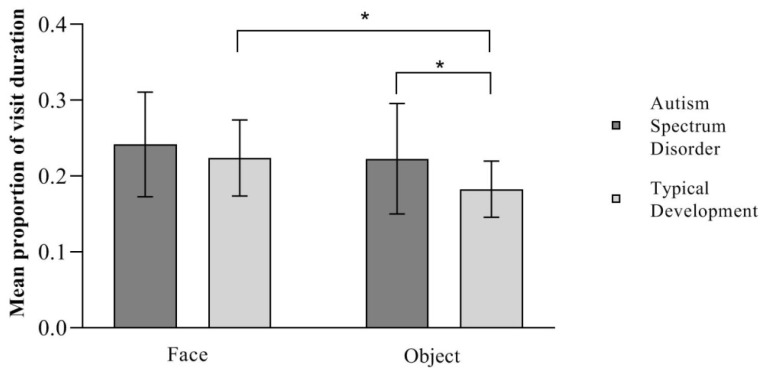
Mean proportion of visit duration towards the stimuli. * *p* < 0.05. Error bars depict a 95% confidence interval.

**Table 1 ijerph-21-00421-t001:** Samples’ characteristics.

	ASD			TD			Statistical Comparison	
	Mean	SD	Range	Mean	SD	Range	*t* (df)	*p*
**Age (months)**	56.63	8.43	45–73	60.42	2.27	57–64	1.893 (20.59)	0.073
**NVIQ** **Leiter-R**	110	13.38	90–133	109	10.69	90–126	−0.244 (34)	0.808

## Data Availability

The data presented in this study are available on request from the corresponding author.

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
