# Peer review of "Attention toward Social and Non-Social Stimuli in Preschool Children with Autism Spectrum Disorder: A Paired Preference Eye-Tracking Study"

_ijerph, 2024, doi:10.3390/ijerph21040421_

Round 1

Reviewer 1 Report

Comments and Suggestions for Authors

The purpose of the study is to examine possible group differences in attention to social and nonsocial stimuli in children with ASD compared to age- and cognitively matched controls in terms of priorities, preferences (in the absence of limited objects of interest), and sustained attention,  when solving the problem of tracking visual preferences using static images of faces and objects that do not belong to limited interests.

The introduction provides a detailed overview of the specific features of the attention of children with ASD, aimed at social and non-social stimuli, describes age-related characteristics, different task settings, stimulus material - dynamic and static images, and different methods of analysis.                  Description of materials and methods:  The sample of children is sufficient - 38 children, 19 children with ASD and the control group with typical developmental. The used stimuli (people's faces (five men and five women) with a neutral facial expression combined with an image of an object on a white background) were aligned according to parameters. The study analyzed three indicators of visual attention to social and nonsocial stimuli: time to first fixation, total duration of visit (total time spent viewing the social and nonsocial stimulus), and the average duration of visits to the social and nonsocial stimulus (the sum of all consecutive fixations in the area of interest).

The results of the study showed that children in the ASD group did not significantly prioritize faces over objects and looked less at the entire stimulus than children with TD. Children in the ASD group showed a statistically significant decrease in attention to social stimuli compared to children with TD. However, the total viewing time showed no differences between TD children and children with ASD in terms of time spent observing the face stimuli. Results indicate an effect of age in the ASD group: age was positively correlated with the proportion of total time spent looking at the face.

In the discussion, the authors note that they found reduced attention to social stimuli in young children with ASD in the absence of increased interest in non-social stimuli and explain this by the existence of different mechanisms underlying attention.

In general, the study is very specific, the task is clearly formulated, and the methods of analysis and processing are adequate. The results fit into the general context of research on the specificity of attention of informants with ASD to social and non-social stimuli.

 Comments:

1) Children in the ASD and control groups differ in age - the average age in the control group is higher, with a larger age range in the ASD group.

2) When describing the results, there is a preliminary analysis, and a subsequent check of the data for normality of distribution. Why couldn't the check be done before data processing?

3) There is insufficient information in the discussion about the mechanisms underlying attention in TD children and children with ASD.

4) Authors noted that child males and females can show differences in circumscribed interest-related objects. Are there any differences between girls and boys in the presented study?

5) Did any children with ASD have eating behavior problems? In this case, the choice of cup and dish as a stimulus cannot be considered successful.

 Technical Note

Table 1. NVIQ – enter designation 

Reviewer 2 Report

Comments and Suggestions for Authors

I very much enjoyed reading the manuscript titled “Attention toward Social and Non-social Stimuli in Preschool 2 Children with Autism Spectrum Disorder: A Paired Preference 3 Eye-tracking Study.”  I have only a few comments.

Lines 48 to 51 – List a couple of references to support the statement

Line 59 – the word “claim” has a negative connotation since it is an assertion that lacks evidence or credibility. I recommend using more appropriate wording.

Line 186 – “Neutral human faces were chosen ….”  How were these stimuli selected?  Was it based on objective ratings? Or simply someone’s opinion?  The authors should address this important issue since much of their findings are based on these stimuli.

Reviewer 3 Report

Comments and Suggestions for Authors

The manuscript is interesting and its topic is consistent with the purposes of the Special Issue of of IJERPH

However, there are some suggestions for minor changes and more substantial conceptual/methodological additions.

Minor revisions

1.      In the full text, substitute the possessive pronoun (“we”, “our results”, “our study”, “we wanted to investigate”, “We found” etc.) with expression such as: current/present  study, results of the study etc.

2.      Even if the referee is not a native speaker, she notices that some sentences are too long, therefore she suggests revising the construction of these sentences:

-          “A recent meta-analysis …non-social elements [3]” (lines-35-39)

-          “Finally …theory [17] (lines 51-56)

-          “Klin and colleagues…did” (lines 79-83)

-          Pierce et al. …social images” (lines 83-90)

-          “The findings…over time” (lines 93-97)

-          “Finally …social interests” (lines 115-120)

3.      Some citations are missing in the text: line 68 (several  studies ?), line 101 (Elsabbagh and coll, year ?),

4.      Ethical questions   (lines 171-177): remove from the text, because it is quoted afterwards (lines 487-489).

5.      Reduce the legend of the figures (remove the comment of the results, for example line 286).

Major revisions

The study is interesting and well presented in its theoretical premises and literature review (introduction).  The procedure, experimental tasks and parameters are well described.

However, it is not easy for the reader to understand the originality of the paradigm that the authors present compared to previous studies.

In other words: if the purpose of the research is to introduce a new paradigm, what does these findings add to previous studies?

1.      (aim and hypothesis) In the referee's opinion, the authors should better explain: a) the theoretical or empirical basis on which they built their original paradigm and b) the implications of the results obtained through this (original/new) paradigm

2.      Discussion of the results: it  is the weakest part of the manuscript. In particular, it is not clear what the implications of the study are (considering the large body of research demonstrating children with ASD's preference for non-social stimuli).

(conclusion) Saying that new studies would be useful is perhaps redundant.

3.      Finally, a section is completely missing in which the authors discuss the limitations of their study (small size  of participants’ sample, both in ASD and TD group; age of children (preschoolers): too old compared to the samples (infants, toddlers) of studies that the authors cite correctly).

Other revisions are detailed in the attached file.

Round 2

Reviewer 3 Report

Comments and Suggestions for Authors

The review appreciates revisions to the paper and all responses to comments